# A Game-Theoretic Approach to Solve Competition between Multi-Type Electric Vehicle Charging and Parking Facilities

Meihui Jiang [1], Tao Chen [2,*], Ciwei Gao [2], Rui Ma [3], Wencong Su [4] and Abdollah Kavousi-Fard [5]

1 Suzhou College of Software Engineering, Southeast University, Suzhou 215123, China
2 Jiangsu Provincial Key Laboratory of Smart Grid Technology and Equipment, Southeast University, Nanjing 210096, China
3 Becom Software Co., Ltd., Beijing 100086, China
4 Department of Electrical Engineering, University of Michigan-Dearborn, Dearborn, MI 48128, USA
5 Department of Electrical and Electronic Engineering, Shiraz University of Technology, Shiraz 71557-13876, Iran
* Correspondence: taoc@seu.edu.cn

**Abstract:** This paper investigates the competition problem between electric vehicle charging and parking desks for different owners using a non-cooperative Bertrand game. There is growing attention on electric vehicles from both policy makers and the public charging service provider, as well as the electric vehicle owners. The interaction between different entities forms a competition (game), especially between multi-type electric vehicle charging and parking facilities. Most of the existing studies on charging platforms are about the optimization of the charging platform scheduling strategy or the game relationship between charging platforms and EV users, but there is a lack of exploration on the revenue game between charging platforms. In this paper, the competitive interactions between different charging decks are studied and analyzed using a general game-theoretic framework, specifically the Nikaido–Isoda solution. In the pricing competition model, the pricing strategies of all players and physical constraints, such as distribution line capacity, are taken into consideration. Through the case studies, it is clearly indicated that the game played between different electric vehicle charging/parking decks will always converge to a Nash equilibrium point. Both charging service providers and customers could benefit from such an open and fully competitive energy service ecosystem, which enhances the overall social welfare.

**Keywords:** electric vehicles; game theory; electricity pricing; distribution network; smart grids

## 1. Introduction

Recently, clean energy technology has been greatly emphasized and has gained huge development, meaning that the internal combustion engine which relies on traditional energy sources has gradually come to an end. Therefore, a new round of transportation electrification is taking place. As an environment-friendly, energy-saving, emission-reducing, clean, and convenient means of transportation, electric vehicles (EVs) are developing rapidly in the current social context of advocating environment protection and climate stabilization [1,2]. Inspired and stimulated by governments' energy conservation policies, EVs can respond to the call and create a huge emerging charging service market [3]. At the same time, the influx of EVs into the market will have a huge impact on the existing utility grid and distribution network [4]. Due to this fact, the current research focus has become when and how to consider the interests of power grids, electric vehicle operators, and EV owners [5,6]. From the perspective of charging network operators, some studies have been conducted by using optimal charging pricing models for public EV charging stations.

Compared with the traditional relationship between cars and gas stations, the relationship between EVs and charging decks is more complicated. The electric vehicle goes to the charging deck for charging; thus, the electric vehicle and the public grid directly

control the charging through the interface of the charging deck [7]. With the increase in the number of EVs and the development of EV charging technology, it is foreseeable that the number of charging/parking decks will also explode. Choosing a suitable charging service platform can effectively alleviate the overflow and charging peaking problem caused by the explosion of EVs [8]. The proper configuration of the charging service infrastructure and technical research of the charging platform can also greatly improve the charging efficiency of EVs [9].

The commonly used game-theoretic relationship between charging decks and EVs needs to consider not only the optimization of charging efficiency, but also the discriminative pricing strategy [10,11]. In general, the revenue of EV charging and parking decks consists of parking fees and charging fees. As the number of EVs and charging decks existing in the market increases, all charging decks in the region compete to maximize their own benefits. Therefore, charging decks often adopt different ways to adjust parking fees and charging fees to achieve different revenue portfolio strategies [12]. Based on their needs, charging decks may use a low-price strategy to increase revenue indirectly by gaining more market share, or use a high-price strategy to increase revenue directly. Each different strategy will have a different impact on the relationship between customers and other decks. Meanwhile, this influence will be fed back to the deck to make strategy adjustments or specific price adjustments to maximize their own profits [13]. When all charging decks and customers in the same region want to maximize their own benefits, the competition between decks will become fierce and the interplay of strategies can become extremely complex [14,15]. With a large number of EVs entering the public charging service market, the revenue maximization problem of parking lots with high electric vehicle penetration is highly variable [16]. By studying the competition between these charging decks, game theory can help them maximize their gains in a much more systematic manner. In a transparent market environment where charging vehicles and public charging facilities are increasingly entering the market, studying the relationship between public charging facilities is conducive to promoting sustainable development progress of all competing parties and the harmony of the EV service market.

Most of the existing research work has focused on issues such as battery technology, peak charging demand mitigation, and power distribution. The work in [17] studied the optimization of the charging peak-to-valley transfer model, and the work in [18] introduced distributed charging control for EVs. Energy trading between smart grids and plug-in electric vehicle (PEV) groups was studied in [19]. A smart Parking Garage EV charging method was proposed in [20], which can significantly reduce power system cost while maintaining reliability. The study in [21] discusses the charge–discharge coordination problem between electric vehicles and the grid, emphasizing the method of the multi-level hierarchical controlled charge–discharge form. In the study of [22], the multi-objective economic–technical–environmental optimization concept of electric vehicle charging and discharging was proposed, and the energy cost of end users in the household microgrid environment was modeled and optimized for the first time. Some game-theoretic approaches to charging EVs on decks are presented in [23] with many competitive features presented. The research in [24] studied the competition between electric vehicles and fuel vehicles in delayed pricing decisions under the constraints of carbon emission reduction policies. Different EV charging scenarios and charging strategies within the business models are mentioned in [25,26]. Based on the game-theoretic approach, the work in [25] studied the problems between capacity configuration, location configuration, construction cost and operation cost of charging stations. The work in [26] examined the economics of EV parking lots with renewable energy generation. The problem of charging time was considered in the research of [27] and the charging speed was improved by improving the transformer. The method of reinforcement learning was used in [28] to develop the problem of suppressing traffic oscillation through the cooperation of networking, electric and automatic vehicles, so as to reduce electric energy. The work in [29] summarized the optimal design of charging

stations using various optimization algorithms, studied issues such as grid-connected, off-grid, and renewable energy combinations, and future development directions.

As mentioned above, in general, most research on EVs has focused on the following:

1.  Technical level: the charging technology of electric vehicles has been studied, including issues such as battery materials, performance, and life aging, as well as issues such as vehicle endurance and generator technology [30,31].
2.  Planning level: issues such as charging pile deployment, electrical infrastructure construction, and vehicle charging scheduling arrangements have been studied.
3.  Economic level: the economic benefits of vehicle-to-grid (V2G) technology, the income of electric vehicle parking lots, and government support subsidies are studied.

Among them, the research on the economic benefits of both EV platforms and users is less than the technical aspects, where the research on the price game between platforms is even scarcer. The research in [32] proposed a model-free structure based on the DRL approach, which learns directly from realistic data to achieve the goal of maximizing the profit of charging stations when the charging time, demand, and other conditions are unknown. Most studies address the interest game between charging platforms and EV users, hoping to obtain the maximum benefit for both, while the study of the interest game between charging platforms is lacking. Therefore, in this paper, we address the economic benefits of electric vehicle charging platforms, study and analyze users with different types of needs, and then propose the game theory principle to model the parking and charging service revenue problem. The price strategy is adjusted through the game principle to maximize the respective benefits of charging platforms.

According to many studies, the relationship between conventional cars and gas stations is very different from the relationship between EVs and charging decks. The research on conventional vehicles' experience cannot be directly applied to the study of the relationship between parking decks and charging vehicles. However, the two relationships have some common features that can help understand the transformation of knowledge in building an attractive and competitive business model. For example, both of them need to consider the price sensitivity of customers. Therefore, our research on the relationship between parking platforms and charging vehicles mainly focuses on the differences in service pricing strategies in the game between different charging platforms and the response of elastic prices among different customer groups.

The main contributions of this paper can be summarized as follows:

1.  The competition and pricing strategy between parking/charging decks is studied with the local charging service platform as the main body to guarantee the optimal pricing strategy at the Nash equilibrium point;
2.  The paper applies the game theory principles to study the competitive relationship among parking platforms in order to maximize revenues, explains the nature of the problem using a special non-cooperative Bertrand game theory model, and provides an effective solution based on Nikaido–Isoda equations;
3.  EVs are divided into three groups according to price sensitivity with the quantified responses of different groups of customers through the experimental data, which simulate the customer behavior when the parking decks adopt different pricing strategies, and we obtain the experimental results for verification.

## 2. Problem Formulation

In this paper, we modeled a deck that provides parking and charging services in a local area based on game theory. In order to maximize its own revenue, each electric vehicle charging deck chooses a corresponding pricing strategy, adjusts and optimizes the response of other parking decks and customers to the strategy, and finally achieves a balanced state in which each local charging/parking deck can maximize its own interests. According to the specific needs of electric vehicle customers in reality, this paper divides users into loyal customers who are not subject to changes in conditions, customers with high price sensitivity, and customers with low price sensitivity. For the problem formulation, the

following general process will be followed. The detailed explanation will be provided in the following sections.

1.  The initial conditions of all parking/charging decks in the game theory model are the same, including electricity costs, parking fees, geographical advantages, etc.
2.  The game theory model uses the relaxation algorithm and Nikaido–Isoda function in the iterative process. The Nikaido–Isoda function is used to iteratively update the pricing strategy until the conditions are met to end the iterative process.
3.  The model will eventually reach an equilibrium point. Under the condition that other parking/charging decks keep their pricing strategies unchanged, no matter how this deck changes its strategy, it cannot continue to improve its own revenue.

Figure 1 shows the hypothesized relationship between different EV decks, the utility company, and EV customers and also indicates the revenue structure of EV decks.

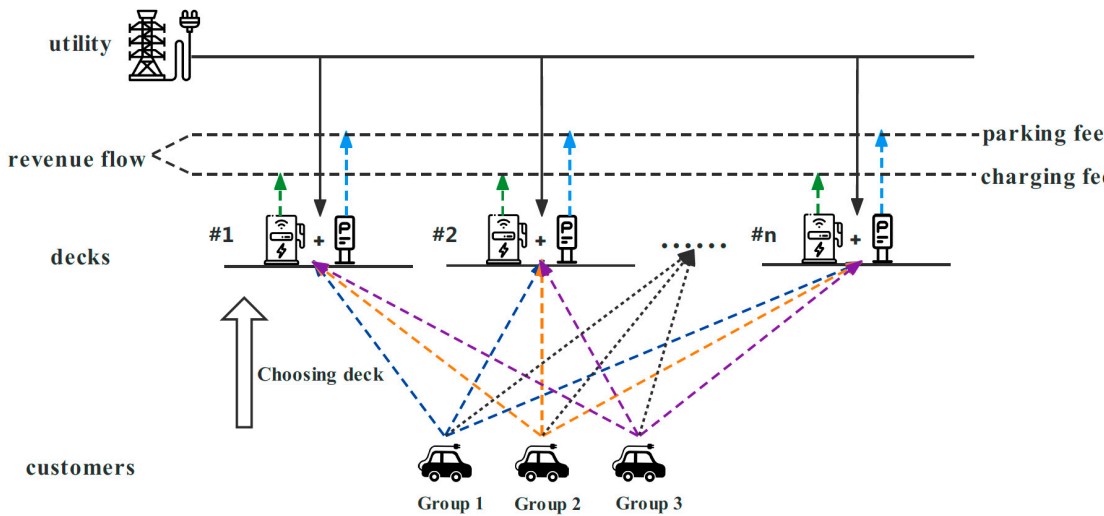

**Figure 1.** The envisioned system architecture of multiple EV charging/parking facilities.

The revenue of the $i_{th}$ EV charging/parking deck consists of two parts, parking income and charging income, which are independent of each other and do not interfere with each other. The goal of the deck is to obtain the maximum revenue, so the goal revenue function for the $i_{th}$ deck is:

$$Max(R_i = R_{ci} + R_{pi} - C_i) \tag{1}$$

Among them, $R_{ci}$ and $R_{pi}$ represent the charging and parking revenue, respectively, $C_i$ represents the cost of each deck per day, thus $R_i$ is the daily net income of the $i_{th}$ deck.

### 2.1. Parking

According to the different needs and preferences of electric vehicle users, all users are divided into three categories of customers; Table 1 describes the groups of customers and gives typical examples of each group of customers.

For the provision of parking services on deck i, the income received can be expressed as follows:

$$R_{pi} = \sum_{t=1}^{24} N_{i,parking} \times \rho_{i,parking} \tag{2}$$

$N_{i,parking}$ represents the number of customers who receive parking service on deck i at time t during time period $\Delta t$.

$$N_{i,parking} = a_{i1}(t) + b_i(t) + c_i(\rho_i, \rho_{-i}, t) \tag{3}$$

where $a_{i1}(t)$, $b_i(t)$, and $c_i(\rho_i, \rho_{-i}, t)$ represent the number of LC, LSC, SC, respectively, who arrive at deck i for parking at time t.

**Table 1.** Customer groups with different attributes.

| Customer Categories | Description | Whether Sensitive to Parking Price | Whether Sensitive to Charging Price | Examples |
|---|---|---|---|---|
| Loyalty customers (LC) | Consists of loyal customers who spend the same amount time charging on a fixed deck each day. | No | No | Customers who live or work locally and do not care about financial expenses. |
| Less-sensitive customers (LSC) | Consists of customers who choose the same deck each day, but whose charging duration varies with the charging price. | Yes | No | Customers who live or work locally, care about financial expenses, and care more about convenience between economy and convenience. |
| Sensitive customers (SC) | Consists of customers who use variable charging decks and have variable charging durations whose choices are influenced by both the price of the $i_{th}$ deck and the average price of all other decks. | Yes | Yes | Passing customers who are willing to spend time on economic planning. |

### 2.2. Charging

With $N_{i,charging}$, we denote the number of customers arriving at the $i_{th}$ deck at time t:

$$N_{i,charging}(t) = a_{i2}(t) + g(\rho_{i,ch}, \rho_{-i,ch}) \times b_i(t) + c_i(\rho_i(t), \rho_{-i}(t)) \tag{4}$$

In the formula, $a_{i2}(t)$ represents the number of LC who arrive at deck i for charging or parking service during time period $\Delta t$. Since LC are not affected by the price, the number of LC who arrive at deck i for charging at time t is only related to the time t.

The charging situation of SC will be restricted by three factors: time t, charging price of deck i, and charging prices of other decks; therefore, the number of SC is expressed as $c_i(\rho_i(t), \rho_{-i}(t))$, which is the same with the number of parking situations. $\rho_i(t)$ means parking and charging fees of the $i_{th}$ deck at time t, and $\rho_{-i}(t)$ represents the average of the sum of charging and parking charges for all other decks at the same time.

The $g(\rho_{i,ch}, \rho_{-i,ch})$ in (4) is used to represent the influence coefficient of the price change of the charging platform on the user, which is negatively correlated with the price. When prices increase, users tend to reduce their charging time requirements, and vice versa. $\rho_{i,ch}$ represents the charging price of the $i_{th}$ deck, and $\rho_{-i,ch}$ represents the average charging price of other decks. Thus, we use $g(\rho_{i,ch}, \rho_{-i,ch})$ to fix the number of LSC charging on the deck.

From the perspective of game theory, we can assume that, in this game, all players (parking/charging decks) satisfy the following properties:

$$\frac{\partial g(\rho_{i,ch}, \rho_{-i,ch})}{\partial \rho_{i,ch}} \leq 0 \tag{5}$$

$$\frac{\partial g(\rho_{i,ch}, \rho_{-i,ch})}{\partial \rho_{-i,ch}} \geq 0 \tag{6}$$

$$\frac{\partial^2 g(\rho_{i,ch}, \rho_{-i,ch})}{\partial \rho_{-i,ch}^2} \leq 0 \tag{7}$$

Equation (5) shows that when the price of charging deck i increases and the prices of other decks remain unchanged, customers are less likely to choose deck i; Equation (6) shows that when the price of charging deck i does not change and the prices of other decks increase, customers are more likely to choose deck i; and Equation (7) shows that when the price increases to a certain extent, this possibility factor will converge to a certain stable value state.

In summary, we define $g\left(\rho_{i,ch}, \rho_{-i,ch}\right)$ as the following expression, as in previous work [16].

$$g\left(\rho_{i,ch}, \rho_{-i,ch}\right) = -\rho_{i,ch}{}^2 + \alpha\rho_{-i,ch} - \frac{\alpha}{4}\rho_{-i,ch}{}^2 \tag{8}$$

where $\alpha$ is a positive number.

Therefore, the amount of charge $D_i(t)$ that deck i needs to provide is:

$$D_i(t) = N_{i,charging}(t) \times P_{charging} \tag{9}$$

The total revenue of the $i_{th}$ deck is expressed as:

$$R_{ci} = \sum\nolimits_{t=1}^{24}\left(\rho_{i,ch} - \rho_{i,buyin}(t) \times D_i(t)\right) \tag{10}$$

where $\rho_{i,buyin}(t)$ represents the price of electricity purchased by deck i and $D_i(t)$ represents the required electricity.

### 2.3. Charging Constraints of Electric Vehicles

This paper does not involve the vehicle-to-grid (V2G) situation, so there is only charging behavior and no discharging behavior. The specific constraints are divided into charging demand constraints, grid dynamic constraints, and electricity price dynamic constraints.

The dynamic constraints of electricity prices have been discussed in the parking and charging part of electric vehicles, so we will discuss other constraints in this section.

### 2.3.1. Constraint of Charging Demand

Due to the limitation of the battery capacity of EVs, the charging demand of EVs should be less than the difference between the maximum capacity of the battery and the capacity of the charging initial state:

$$SOC_{dem} \leq SOC_{max} - SOC_{ini} \tag{11}$$

where $SOC_{dem}$ represents the charging demand of EVs, $SOC_{max}$ represents the upper limit of EV battery charging, and $SOC_{ini}$ represents battery level before charging.

$$SOC_{dem} = \frac{\Delta T \sum p_t^{chg} s_t^{chg} \eta_{EV}^{chg}}{CAP_{EV}} \tag{12}$$

where $s_t^{chg}$ represents the charging state of EV in time period t, $p_t^{chg}$ represents charging power, $\eta_{EV}^{chg}$ represents charging efficiency, and $CAP_{EV}$ represents the capacity of the EV battery.

$$s_t^{chg} = \begin{cases} 1, & \textit{EV charges in time period t} \\ 0, & \textit{EV does not charge in time period t} \end{cases} \tag{13}$$

### 2.3.2. Constraint of Dynamic Grid

Due to the randomness of the charging behavior of EVs, when large-scale electric vehicles are connected to the grid at the same time, it will bring burden to the local distribution network. Therefore, safe operation of the grid at limited charging levels needs to take into account the dynamic constraints of the grid, including transformer and line constraints, phase unbalance, and voltage stability within the network. Generally, system constraints on charging vehicles are mainly in terms of transformer capacity, line current, voltage drop, and phase offset [33].

The specific constraints are expressed as follows:

$$V_{Tx}x_{\varnothing,t} \leq \frac{1}{3}P_{Tx}^{max} \times 130\%, \varnothing \in \{A, B, C\} \tag{14}$$

$$x_{\varnothing,t} \leq x_{\varnothing}^{\max}, \varnothing \in \{A, B, C\} \tag{15}$$

$$x_{k,t}^{\max} \leq x_k^{\max} \tag{16}$$

where $V_{Tx}$ represents distribution transformer source voltage, $x_{\varnothing,t}$ represents total current generated by all single-phase loads on the phase, $P_{Tx}^{\max}$ represents transformer rated power, $x_{\varnothing}^{\max}$ represents backbone cable rated current, and $x_k^{\max}$ represents service line rated current.

In Equations (15) and (16), because the backbone and service lines usually have different specifications, we introduce separate constraints for each stage of the backbone network.

$$V_{h,t}^{drop} = \sum_{j=1}^{h} \left( I_j \sum_{k=1}^{j} z_{k-1, k} \right) \tag{17}$$

where $V_{h,t}^{drop}$ represents the difference between transformer voltage and charging station voltage.

$$V_{Tx} - V_{h,t}^{drop} > V^{min} \tag{18}$$

Equation (18) ensures that the voltage of each line can be maintained at a normal level, not lower than the minimum value for normal operation.

To keep our constraint set linear, we express the phase imbalance as a percentage deviation from the average phase loading:

$$\frac{\left| x_{\varnothing,t} - \frac{1}{3}(x_{A,t} + x_{B,t} + x_{C,t}) \right|}{\frac{1}{3}(x_{A,t} + x_{B,t} + x_{C,t})} < p, \varnothing \in \{A, B, C\} \tag{19}$$

## 3. Game Theory and Solution

### 3.1. Game Theory Definition and Concepts

This section describes the game theory algorithm used to solve the problem. Based on the game theory, this paper treats each deck as a player, and all decks together participate in a competitive game in order to maximize revenue. We set X to represent the strategies of all decks, in which each vector $X_i$ represents the strategy adopted by deck i, set f represents the payoff of all decks, and each element $f_i$ represents the payoff of the $i_{th}$ deck when all decks correspond to strategy X. When $(y_i|x)$, it means that when other decks adopt the $(x_1, \ldots, x_{i-1}, y_i, x_{i+1}, \ldots, x_I)$ strategy set, the $i_{th}$ deck adopts the strategy of $y_i$, so according to the definition of the Nash equilibrium:

$$x^* = (x_1^*, \ldots, x_I^*), \forall i \tag{20}$$

$$f_i(x^*) = \max_{(X_i|x^*) \in \rho} f_i(X_i|x) \tag{21}$$

$$X = X_1 \times X_2 \times \ldots \times X_I \tag{22}$$

Therefore, when the competition for maximum revenue among all decks reaches a Nash equilibrium, the strategies adopted by all decks satisfy the following conditions:

$$f_i\left(x_i^*|x^*\right) \geq f_i\left(x_i|x^*\right), \forall i \tag{23}$$

It can be seen from formula (13) that a single deck cannot achieve an increase in revenue when other decks do not change their strategies, no matter how it changes its own strategies.

### 3.2. Nikaido–Isoda Function

The search problem of the Nash equilibrium point can be transformed into the optimization problem of the Nikaido–Isoda function for computational solutions [34]. The

point of the Nash equilibrium could be found by the Nikaido–Isoda iterative process. The Nikaido–Isoda function can be expressed as:

$$F(x, y) = \sum_{i=1}^{I} [f_i(y_i | x) - f_i(x)] \tag{24}$$

Before the game reaches the equilibrium point, when the strategy of a deck changes from $x_i$ to $y_i$ while the strategies of all remaining decks remain unchanged, the revenue of deck i can be improved. Therefore, $F(x, y)$ represents the sum of the gains from unilateral strategy changes across all decks.

The Nash equilibrium is reached when the total increase in revenue of all decks unilaterally adjusting their strategies is 0, that is, no deck can achieve an increase in revenue by merely adjusting its own strategy.

$$\text{Max}_{(x*y) \in X} F(x^*, y) = 0 \tag{25}$$

Reaching Nash equilibrium requires that $g(\rho_{i,ch}, \rho_{-i,ch})$ is a convex and concave function in the case of a concave function, so we need to ensure that the conditions are satisfied [35]:

$$Z(x) = \text{argmax}_{y \in X} F(x, y) \ x, \ Z(x) \in X \tag{26}$$

### 3.3. Relaxation Algorithm

The expression of the relaxation algorithm is as follows:

$$x^{s+1} = (1 - \alpha_k) x^s + \alpha_k Z(x^s) \tag{27}$$

where $0 < \alpha_k < 1$.

In the formula, the iteration of the $(s + 1)_{th}$ step consists of the current point $x^s$ and the $Z(x)$ value of the $s_{th}$ step. Such expressions ensure that the algorithm can converge under certain conditions [36,37]. Since we only know the past strategy and revenues of each deck, the information on the equilibrium point cannot be directly obtained through the relaxation algorithm. Therefore, we assume that the principle of strategy selection is the same for each deck and does not change.

In order to simplify the calculation process, we set the value of $x^s$ to 0.5. In this way, it converges to an equilibrium point when enough iterations are made. The number of iterations depends on the convergence precision we set. The conditions for stopping the iteration are as follows [38]:

$$\text{Max}_{(x^s,y) \in X} \Psi(x^s, y) < \delta \tag{28}$$

$\delta$ is an extremely small number we use to control the precision.

## 4. Numerical Results

### 4.1. Experiment Settings and Condition Configuration

In the simulated system settings, in order to exclude the influence of non-experimental factors, it is assumed that all goals are to maximize their own benefits. The unit power cost per charging/parking deck is the same as the operating cost. In addition, all decks are not restricted by other external factors when making strategic adjustments. That is, all decks independently adjust electricity prices and parking fees with the goal of maximizing profits.

We set up six decks owned by six different subjects in the system, and the parameters in Table 1 show parameters of LSC and SC customer weighting function $g(\rho_{i,ch}, \rho_{-i,ch})$, charging power, start point, and tolerance.

The experiment set a minimum time span of 1 h and studied the hourly arrivals of EVs on decks to receive parking or charging services (24 h a day). Different prices can be set for each deck at each hour of a day (up to 24 different price strategies can be set). The process of making price decisions in the system based on game theory is shown in Figure 2.

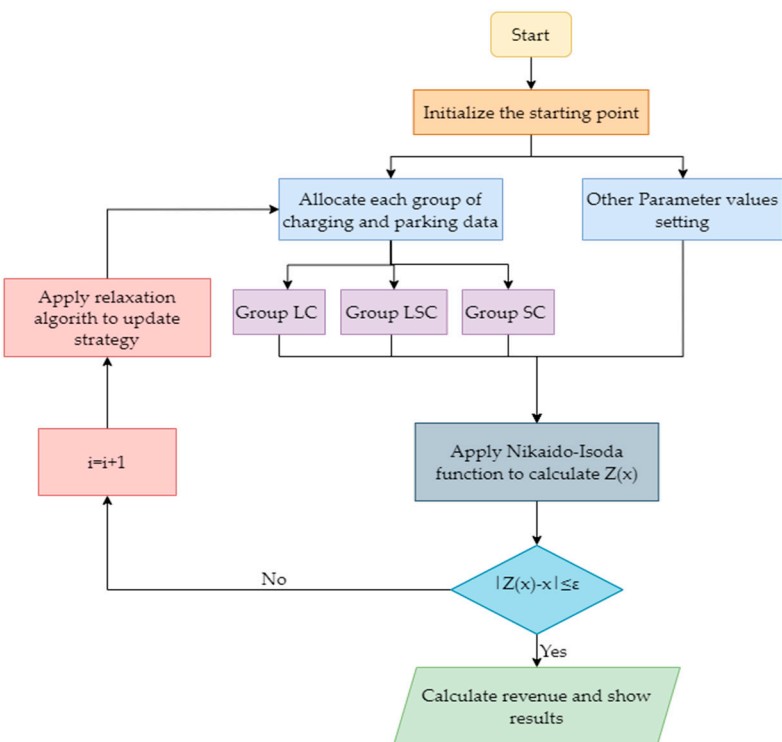

**Figure 2.** Flowchart of the proposed game-theoretic algorithm.

At the beginning of the competition, we set all decks to the same initial conditions. In the process of adjusting the price strategies, we used the Nikaido–Isoda function and relaxation algorithm for calculation to find the Nash equilibrium point. The iteration is ended when the difference between the impact of the next decision on the revenue and the impact of the current decision is greater than the coefficient of ending the iteration. Otherwise, it goes to the next iteration.

In the following experiments, we first studied the convergence of the pricing of each deck in the game to verify that the game can reach the Nash equilibrium point. Second, we studied the effect of operating conditions between decks on the price decision and the impact on the earnings of other groups of decks. Finally, we investigated the impact of different customers on the pricing strategy and revenue of decks, while changing the weight of different customer groups by keeping the total number of customers on each deck constant.

### 4.2. Case Study

#### 4.2.1. Check of Convergence

We assume that the six local decks are owned by six different subjects that are only in competition with each other. Therefore, the goal of all decks is to maximize their own benefits under the same environmental background and resource constraints. In the same area, the number of electric vehicle customers remains basically the same, but the price strategy of each deck will be different because the decks will have different operating conditions in different time periods.

We set the same charging price and parking fee for all decks. When the iteration starts, each deck adjusts according to its own price strategy, and the price changes begin to differ from each other. In Figure 3a,b, we can see that as the number of iterations increases, the charging prices and parking fees for all decks eventually converge around a certain value. We control the number of iterations by setting the precision that satisfies the iteration stop. When the iteration is stopped, the system reaches a Nash equilibrium. At this point, neither deck can improve its own income under the condition that the strategies of other decks remain unchanged.

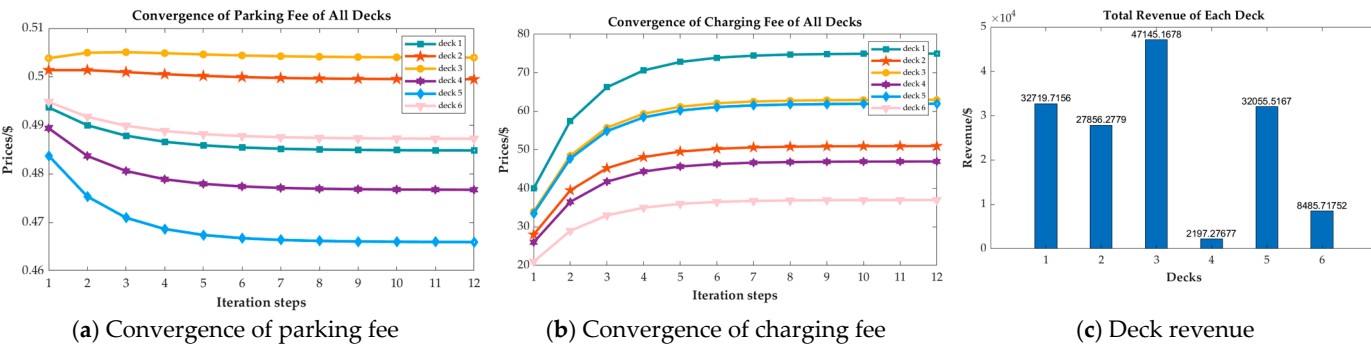

**Figure 3.** The convergence of parking/charging fees of all decks and deck revenue.

In Figure 4, the revenues of the six decks vary widely due to the differences in parking fees and electricity prices. Decks 1, 4, 5, and 6 adopt the strategy of reducing the charging electricity price, and decks 2 and 3 adopt the strategy of increasing the charging electricity price. All decks have a strategy of raising parking fees, but decks 1, 3, and 5 have a larger price increase. Under different strategies, the final revenue of each deck is shown in Figure 3c. Deck 3 obtains the highest payoff and deck 4 obtains the lowest payoff.

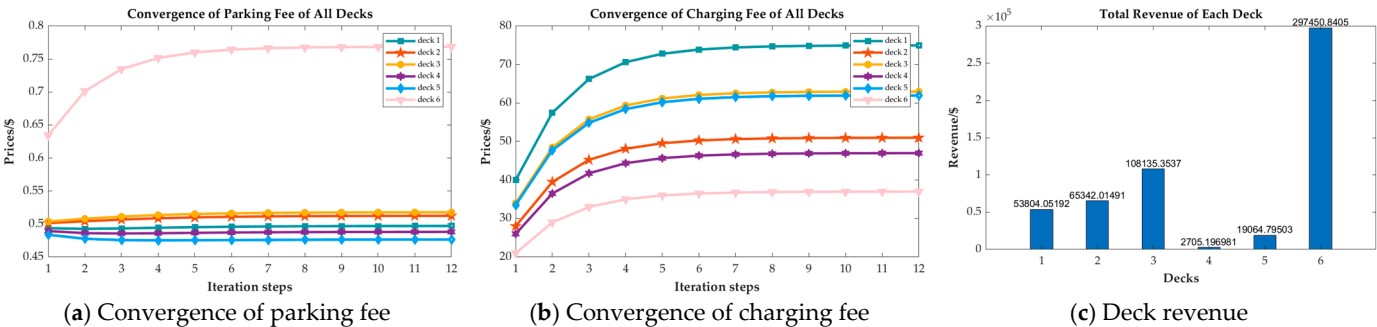

**Figure 4.** The charging price, parking fee, and total revenue after the number of SC customers in deck 6 was reduced to zero.

### 4.2.2. Evaluate Customer Influences

Since the direct source of the deck's income is the customer, whether or not the customer chooses to park or charge on a certain deck is very important to the deck's income. As described in Table 1, this paper divides all EV users into three groups according to their needs. This section mainly examines the impact of different pricing strategies adopted by the decks on customers and other decks. Feeding this influence back to the player side enables decks to have more information to refer to when making pricing decisions.

In order to study the operating conditions of one of the decks (here measured by the number of vehicles parked and charged on that deck), we reduced the number of SC customers in that deck to zero for the most intuitive and obvious effect. We observe the changes in the number of customers and pricing strategies of other groups of decks when the number of SC customers in this group of decks reaches zero. Figure 4 shows how the pricing strategies of all decks have changed.

From Figure 5, we can see that when there are no price-sensitive customers in a deck, the decks tend to take higher prices, and a change in this behavior does not affect the pricing of the deck with the same composition of other customers.

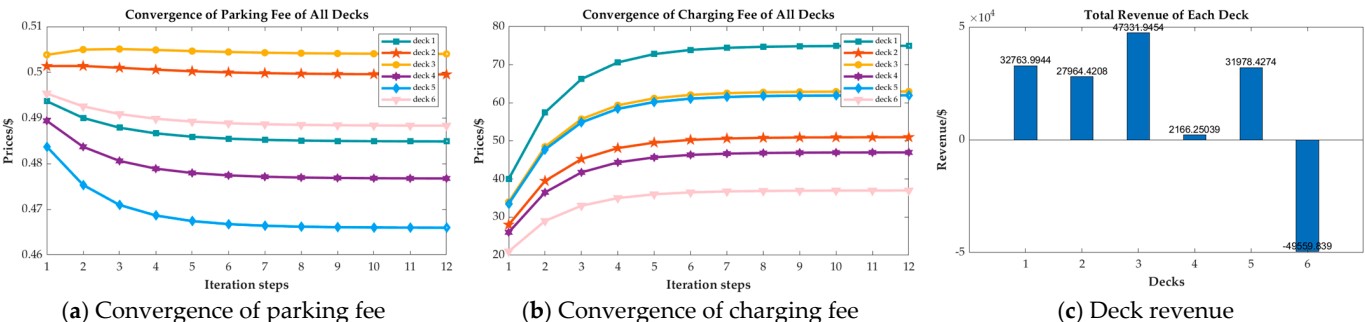

(**a**) Convergence of parking fee     (**b**) Convergence of charging fee     (**c**) Deck revenue

**Figure 5.** The charging price, parking fee, and total revenue after the number of LSC customers in deck 6 was reduced to zero.

In the above case, we studied the impact of changes in customers of one deck on the pricing strategies and revenues of other decks. Next, we studied the impact of pricing strategies on price-sensitive customers. We reduced all decks by 100 SC users and added them as a combination of LC and LSC customers to each deck. Under these conditions, we can find that each deck will tend to incur higher parking and charging fees. This is not difficult to understand—the decks do not need to consider that they will lose customers if they raise prices. Therefore, they can use higher prices in order to obtain higher returns. This is consistent with the facts.

As can be seen from Figures 4–8, SC customers are more sensitive to price; therefore, the number of SC customers has a greater impact on the pricing of decks. Generally, a large change in the number of customers in a deck will affect the price fluctuations of that deck, whereas changes in the number of customers in the LSC and LC groups have less impact on the pricing of decks. However, because the number of LSC customers accounts for the majority of the total number of customers in a deck, the loss of LSC customers will cause huge losses to the deck.

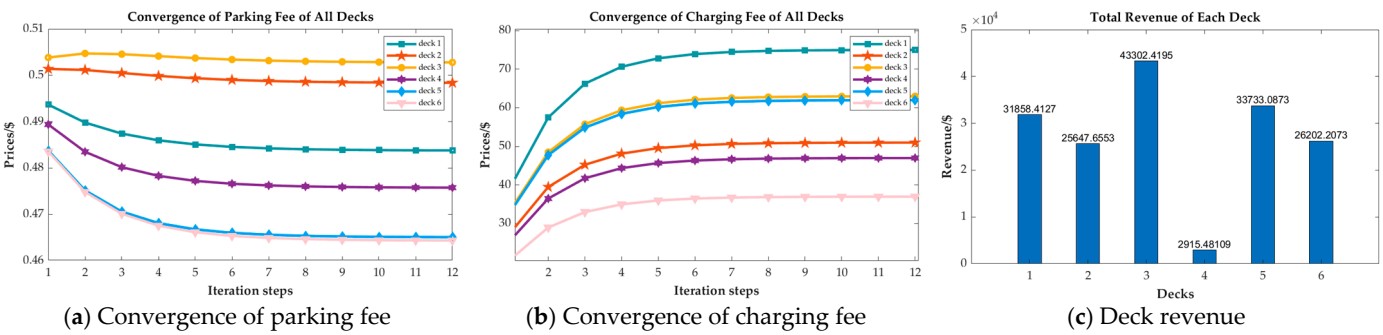

(**a**) Convergence of parking fee     (**b**) Convergence of charging fee     (**c**) Deck revenue

**Figure 6.** The charging price, parking fee, and total revenue after the number of LC customers in deck 6 was reduced to zero.

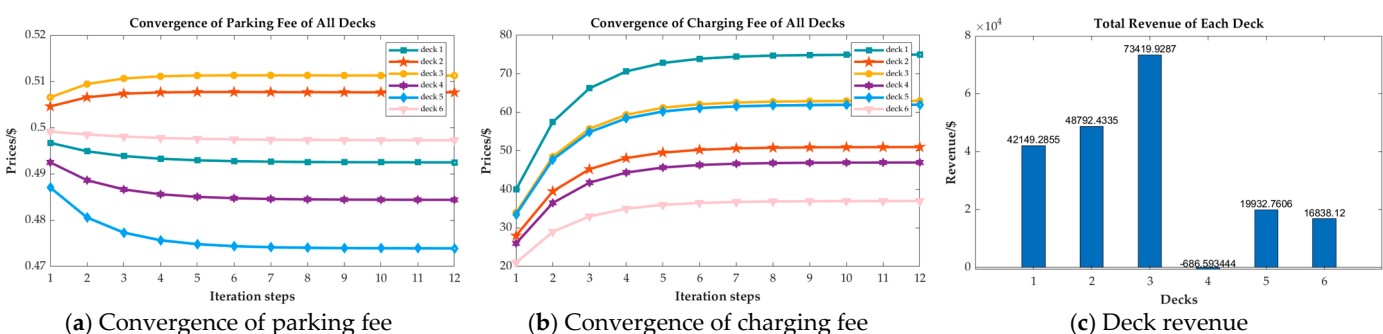

(**a**) Convergence of parking fee     (**b**) Convergence of charging fee     (**c**) Deck revenue

**Figure 7.** The electricity price, parking fee, and total revenue after halving the number of SC customers in all decks and including them in Group LC.

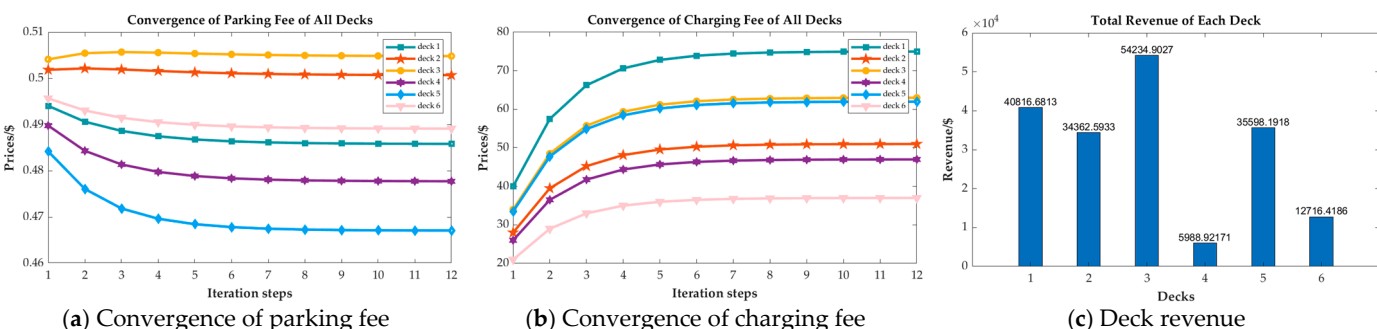

**Figure 8.** The electricity price, parking fee, and total revenue after halving the number of SC customers in all decks and including them in Group LSC.

Then, we removed all LC customers in all decks and added them to Group LSC, or halved the number of LSC customers and added them to Group LC to further investigate the impact of changing customer weighting. Figure 9 shows the electricity price, parking fee, and total revenue after reducing all LC customers in all decks and including them in Group LSC, while Figure 10 shows the electricity price, parking fee, and total revenue after reducing half of all LSC customers in all decks and adding them to Group LC. More or less, there is an increasing trend in revenues for both deck 1 and deck 6 in these two situations. For decks 2 and 3, their revenues tend to increase when the number of LC customers increases, and decrease as the number of LC customers decreases. For decks 4 and 5, their revenues tend to increase when the number of LSC customers increases, and decrease as the number of LSC customers decreases.

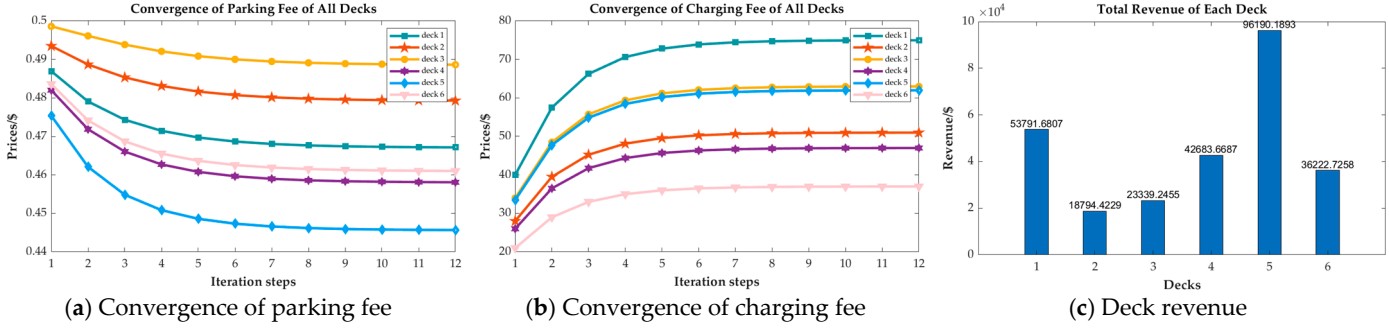

**Figure 9.** The electricity price, parking fee, and total revenue after reducing all LC customers in all decks and including them in Group LSC.

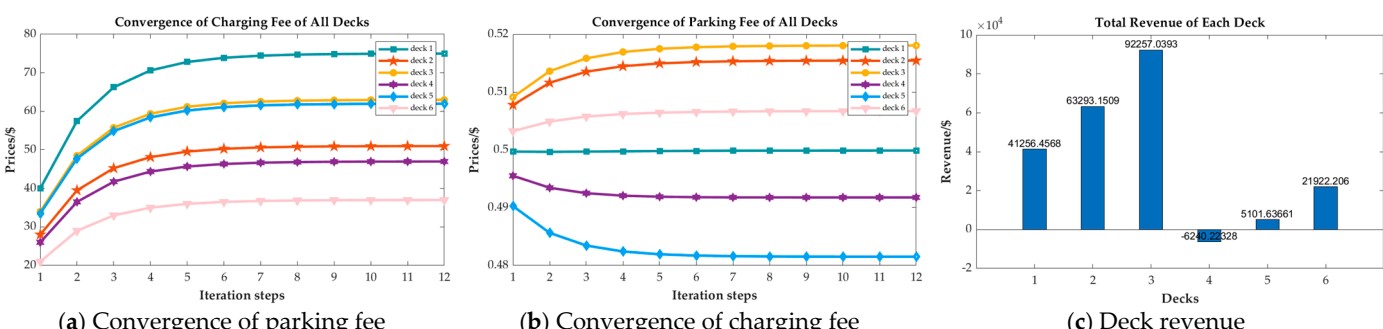

**Figure 10.** The electricity price, parking fee, and total revenue after halving the number of LSC customers in all decks and adding them to Group LC.

In this case, Group SC customers, who obtain the highest level of flexibility, have the most influence on pricing strategies of all decks. The changes/number of LC customers and LSC customers in one deck have limited influence on pricing strategies of other decks.

However, the change of numbers in these customers of one deck has considerable influence on the changes in Group SC customers. The influence that different customer groups have on the decks' revenue varies a lot. Decks 1, 2, 3, and 6 rely more on Group LC customers, while decks 4 and 5 rely more on Group LSC customers. In the real world, deck owners may apply rebate policies or other benefit policies in order to attract different kinds of customers, improving their revenue in other ways on top of changing their pricing strategies.

According to the study, it can be concluded that the LSC customer group has the largest number of EV customers and has the greatest impact on the revenue of decks. When a specific pricing strategy of a certain EV charging service deck cannot attract enough customers from the LSC group, the competence among different charging decks will become more intensive. Additionally, Group SC customers are extremely sensitive to the charging price, and the number of such customers is quite small. Considering this, it is actually wise for deck owners to first reduce parking fees to attract more types of incoming customers and later increase charging prices to increase revenues if they find that there are fewer SC customers and more LSC and LC customers in the deck.

## 5. Conclusions

This paper studies the revenue maximization problem of the EV charging service based on a game theory model. The model transforms the problem into finding the Nash equilibrium point through relaxation function iteration, and solving the problem using the Nikaido–Isoda relaxation algorithm. The increased revenue is obtained by adopting different pricing strategies among multiple decks in a specific area and in an open full competitive market environment. The simulation results demonstrate that each player (i.e., charging deck) in the game model can reach the Nash equilibrium and achieve its own best pricing strategy at the same time. In addition to studying the impact of the game formed by the competitive relationship between multiple charging posts in the same region on the charging posts' own pricing strategies, this paper also investigates the impact of EV customers with different sensitivities on charging pricing and ultimate revenue. In summary, the revenue maximization game model of the charging service studied in this paper can help promote the optimization of the fair competition in the EV charging/parking service market with a suitable business model being designed. In the paper, the changes in customer demand, dynamic tariffs, real-time grid conditions constraints, etc., have not yet been developed and studied in depth, which will be the next step in the work.

**Author Contributions:** Conceptualization, T.C.; Methodology, M.J. and W.S.; Software, M.J.; Investigation, W.S. and A.K.-F.; Data curation, R.M.; Writing—original draft, M.J.; Writing—review & editing, T.C., C.G., R.M. and A.K.-F.; Supervision, T.C. and C.G.; Project administration, C.G. and W.S.; Funding acquisition, T.C. All authors have read and agreed to the published version of the manuscript.

**Funding:** This research was funded by National Natural Science Foundation of China under grant number:52107079, Natural Science Foundation of Jiangsu Province under grant number: BK20210243; and the Open Research Project Programme of the State Key Laboratory of Internet of Things for Smart City (University of Macau) under grant number: SKL-IoTSC(UM)-2021-2023/ORPF/A14/2022.

**Data Availability Statement:** The data presented in this study are available on request from the corresponding author.

**Conflicts of Interest:** The authors declare no conflict of interest.

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
