# Peer review of "A Game-Theoretic Approach to Solve Competition between Multi-Type Electric Vehicle Charging and Parking Facilities"

_wevj, doi:10.3390/wevj14030059_

Round 1
Reviewer 1 Report
Authors have presented interesting study, which covers management of EV charging using Game theory approach.
1. The state-of-art literature review should be updated for identifying the key research objectives of the presented work.
2. Authors should consider the energy requirements of the parked electric vehicles, users' preferences based on charging pricing mechanisms for doing the changing management of plugged electric vehicles at parking place.
3. Authors are advised to include dynamics of the grid constraints in the time varying charging management strategies.
4. Dynamics in electrical vehicle charging pricing should be analyzed. There should be some sensitivity analysis on EV charging prices and the EV demand variations considering the EV users preferences.
5. Results and discussion, conclusion, abstract should be revised considering above mentioned points.
Author Response
The authors would like to thank the comments from reviewers and pointing out the multiple issues that should be addressed. In the revision, we have carefully assessed all the comments from reviewers and addressed all of them in details with clear marking in the manuscript.

Reviewer 2 Report
This paper presents a game theoretic approach (based on the Bertrand game specifically) to solve the competition among vehicle owners in multi-type charging stations. The paper is interesting and the topic timely and suitable for this journal. Please consider my comments below:
- English must be polished. I recommend revision by a native speaker.
- Literature Review is very narrow. Some articles have been presented recently related to optimal management of multi-type charging stations (e.g. 10.1016/j.energy.2022.124219). The authors should carry out a more comprehensive analysis of existing works, in order to better justify the contributions of this paper.
- In general, the methodology developed in this paper is not well explained and the overall paper may be difficult to follow. I recommend describing the methods in a more comprehensive way.
- In the results section, the authors should include a scalability analysis, showing how the size of the problem affects to the computational performance of the method proposed. Note that this may be a critical point for the practical implantation of this methodology in industry tools.
- Note that there are two fourth sections, please fix it.
Thanks to the authors for your effort and time.
Author Response

(The authors gave the same response as above.)

Reviewer 3 Report
In this paper the authors analyze the problem of competition among different EV charging and parking decks. Based of noncooperative Bertrand game the dependences between multi-type EV charging and parking facilities are modeled. In the proposed game-theoretic framework, EV charging facility plays an important role in deciding its price to obtain the maximal revenue under specific constraints. The competitive interactions among different charging decks are studied using a general game-theoretic framework and particularly Nikaido-Isoda solutions. Considering that all decks independently adjust electricity prices and parking fees with the goal of maximizing profits, the simulation results demonstrate each player charging deck in the game model can reach the Nash equilibrium and achieve its own best pricing strategy at the same time.
The proposed method is well conducted and presented.
Comments to authors:
1. In the Introduction section the current literature review can be improved. For example you can examine the merit and demerit of the other existing studies: i) Advanced software system for optimization of car parking services in urban area, DOI: 10.1109/ATEE.2013.6563510; ii) Harmonic distortion caused by EV battery chargers in the distribution systems network and its remedy, DOI:10.21427/K5QD-ST78; iii) Research on electric vehicle charging system: key technologies, communication techniques, control strategies and standards, DOI: 10.1109/ICEPES52894.2021.9699496
2. Please define:
- the acronym PEV page 4/line 137;
- the variables LC, LSC, SC used in page 4/line 151;
- e.a.
3. In my opinion, in addition to function Max (1)- page 4/line 137, another complementary function Min can be defined, which minimizes the cost (or time) of moving the EV to the charging point, having as input variable the actual state of charge of the battery.
Author Response

(The authors gave the same response as above.)

Round 2
Reviewer 2 Report
Dear authors,
I have no more comments.